# Savory, Oregano and Thyme Essential Oil Mixture (HerbELICO^®^) Counteracts *Helicobacter pylori*

**DOI:** 10.3390/molecules28052138

**Published:** 2023-02-24

**Authors:** Ivan Nikolić, Eng Guan Chua, Alfred Chin Yen Tay, Aleksandra Kostrešević, Bojan Pavlović, Katarina Jončić Savić

**Affiliations:** 1Oncology Institute of Vojvodina, Faculty of Medicine Novi Sad, University of Novi Sad, Put Doktora Goldmana 4, 21204 Sremska Kamenica, Serbia; 2Helicobacter Research Laboratory, The Marshall Centre for Infectious Diseases Research and Training, University of Western Australia, Perth, WA 6009, Australia; 3Polarislab, Branka Bajića 78, 21000 Novi Sad, Serbia; 4Phytonet Ltd., Science Technology Park, Veljka Dugoševića 54, 11000 Belgrade, Serbia

**Keywords:** *Helicobacter pylori*, essential oils, *Satureja* L., *Origanum* L., *Thymus* L., dietary supplement

## Abstract

Fifty percent of the world’s population is infected with *Helicobacter pylori,* which can trigger many gastrointestinal disorders. *H. pylori* eradication therapy consists of two to three antimicrobial medicinal products, but they exhibit limited efficacy and may cause adverse side effects. Alternative therapies are urgent. It was assumed that an essential oil mixture, obtained from species from genera *Satureja* L., *Origanum* L. and *Thymus* L. and called the HerbELICO^®^ essential oil mixture, could be useful in *H. pylori* infection treatment. HerbELICO^®^ was analyzed by GC-MS and assessed in vitro against twenty *H. pylori* clinical strains isolated from patients of different geographical origins and with different antimicrobial medicinal products resistance profiles, and for its ability to penetrate the artificial mucin barrier. A customer case study included 15 users of HerbELICO^®^  *liquid*/HerbELICO^®^  *solid* dietary supplements (capsulated HerbELICO^®^ mixture in liquid/solid form). Carvacrol and thymol were the most dominant compounds (47.44% and 11.62%, respectively), together with *p*-cymene (13.35%) and *γ*-terpinene (18.20%). The minimum concentration required to inhibit in vitro *H. pylori* growth by HerbELICO^®^ was 4–5% (*v*/*v*); 10 min exposure to HerbELICO^®^ was enough to kill off the examined *H. pylori* strains, while HerbELICO^®^ was able to penetrate through mucin. A high eradication rate (up to 90%) and acceptance by consumers was observed.

## 1. Introduction

*Helicobacter pylori* is a gram-negative, microaerophilic, spiral-shaped bacterium that inhabits the human stomach, causing several severe gastrointestinal disorders including acute and chronic gastritis, peptic and duodenal ulcers, gastric adenocarcinoma and gastric mucosa-associated lymphoid-tissue (MALT) lymphoma [1,2]. It was indicated that approximately 50% of the world’s population is infected with *H. pylori*, with prevalence ranging from 25% to 50% in developed nations and from 70% to 90% in developing countries [2,3].

Although most of the infected people never experience symptoms of disease, there is a risk that some of them may develop a peptic ulcer (approximately 10 to 20% of infected subjects), one quarter (approximately 4.25%) may have serious ulcer complications, while 1 to 2% may progress to gastric cancer [2]. Therefore, the efficient eradication of *H. pylori* from gastric mucosa is a necessity.

Current treatment regimens for *H. pylori* infection usually consist of a proton pump inhibitor (PPI) and two different antimicrobial medicinal products among amoxicillin, clarithromycin, metronidazole, levofloxacin, rifabutin, furazolidone and tetracycline [4]. However, due to the increased rate of antibiotics and other antimicrobial medicinal products resistance in *H. pylori*, the treatment success rate is declining. In addition, many of these antimicrobial medicinal products are often associated with adverse side effects such as drug allergy, diarrhea, nausea, vomiting, bloating and abdominal pain, leading to increased numbers of noncompliant patients [3,4]. Thus, there is an urgent need for a novel therapy which is effective against *H. pylori* infection and at the same time has few side effects.

While several alternative *H. pylori* eradication therapies [5] based on microorganism products [6], peptides [7,8], polysaccharides [9] and intragastric violet light [10] were not effective in eradicating the bacteria, they could lower the bacterial load and are therefore suggested to be used in combination with antibiotics or other antimicrobial medicinal products to reduce the side effects. Traditional medicinal plants are also commonly used for the treatment of gastrointestinal disorders, including *H. pylori* infection [11]. The proposed mechanisms underlying the anti-*H. pylori* activity of natural products include the disruption of the bacterial membrane, interference with integral membrane proteins, urease inhibition, DNA damage and protein synthesis inhibition [11,12,13,14]. In vitro and in vivo studies have demonstrated that some aromatic plant-derived essential oils and their components exhibit antimicrobial activity against *H. pylori* [15,16,17,18,19,20,21,22]. For example, oregano [17,21] and thyme [17,18,21] essential oils, as well as their major components carvacrol [17,22] and thymol [17,21], had in vitro anti-*H. pylori* activity. The results of these studies, commonly expressed as minimal bactericidal concentrations (MBC) or minimal inhibitory concentration (MIC), implicate that the levels of their activities vary to some extent, but also evidence some synergistic effects [22]. For example, the observed inhibition zone of *H. pylori* growth by oregano and thyme oils were 1.9 ± 0.4 and 1.5 ± 0.5 cm, respectively, while the determined MBC (in a liquid medium) of both samples was 1.0 ± 0.04 g/L [17]. In other studies, the MIC of oregano and thyme oils were 31.3 and 15.6 mg/L [21], respectively, while thyme oil at MIC = 5.4 μg/mL inhibited the growth of *H. pylori* [18]. Carvacrol showed anti-*H. pylori* activity at MBC = 0.04 g/L [17] or MIC = 0.13 mg/mL [22], while the MBC of thymol was found to be 0.1 g/L [17] and 31.3 mg/L (MIC = 7.8 mg/L) [21]. However, the results of different studies usually cannot be compared since different assays are used. In the study of Lesjak et al. [22], savory and oregano essential oils expressed the highest activity (MIC = 2 μL/mL) among nine tested essential oils, but their binary and ternary mixtures were more powerful (MIC ≤ 2 μL/mL), with the most prominent *S. hortensis* and *O. vulgare* subsp. *hirtum* mixture (volume ratio 2:1), which expressed four times higher activity than individual oils (MIC = 0.5 μL/mL) [22]. To the best of our knowledge, this is a unique report [22] on the anti-*H. pylori* activity of a mixture of several essential oils.

So, all of these data may implicate the possibility for inclusion of essential oils in *H. pylori* eradication therapy as safe, natural agents. Many essential oils are generally recognized as safe (GRAS) by the U.S. Food and Drug Administration [23,24]. However, when used undiluted, essential oils can often cause irritation to the oral and gastric mucosa [25]. However, due to their unique physicochemical properties, it is difficult to develop a stable essential oil formulation of pharmaceutical grade, which limits their use in the pharmaceutical industry. To the best of our knowledge, there is currently no commercially available pharmaceutical essential oil product that has been tested in humans for the eradication of *H. pylori* infection.

Therefore, in the present study, we evaluated the HerbELICO^®^ essential oil mixture, a formulation consisting of three different types of essential oils derived from *Satureja* L., *Origanum* L. and *Thymus* L. plant genera, for anti-*H. pylori* activity and its treatment effect in *H. pylori*-positive consumers.

## 2. Results

### 2.1. Chemical Composition of HerbELICO Essential Oil Mixture

A total of 20 compounds were identified in the HerbELICO^®^ essential oil mixture via GC-MS analyses (Table 1, Figure 1). The monoterpenoid phenols carvacrol and thymol were among the most dominant compounds (47.44% and 11.62%, respectively), together with hydrocarbon monoterpenes *p*-cymene (13.35%) and *γ*-terpinene (18.20%). *α*-Terpinen (1.51%), linalool (1.45%) and *β*-caryophyllene (1.56%) were also present in considerable amounts. The rest of the compounds were present in less than 1%.

### 2.2. Determination of the MIC of HerbELICO^®^ Essential Oil Mixture

Most of the *H. pylori* strains that were used in the assays were resistant to multiple antimicrobial medicinal products, which was confirmed by an E-test (Table 2). The HerbELICO^®^ essential oil mixture was tested against 20 *H. pylori* clinical strains (Table 3). The minimum inhibitory concentration (MIC) of the HerbELICO^®^ essential oil mixture was determined using the agar dilution method. The MIC was determined as the lowest concentration of the composition required to completely inhibit *H. pylori* growth (Table 3). The results showed that the MIC required to inhibit *H. pylori* growth, regardless of the strain origin or antimicrobial medicinal products resistance profile, was around 36.6–45.7 g/L, which corresponds to 4–5% (*v*/*v*) since the density of the oil mixture was 0.915 g/mL.

### 2.3. Time-Kill Assay

In this experiment, four *H. pylori* strains were exposed to a 5% HerbELICO^®^ essential oil mixture for 10, 20 and 45 min. The results of this experiment showed that a short 10 min exposure to the HerbELICO^®^ mixture was sufficient to inactivate all *H. pylori* cells (Figure 2).

The untreated group had no exposure to the HerbELICO^®^ essential oil mixture. The treated group had exposure to the 5% (*v*/*v*) HerbELICO^®^ essential oil mixture for 10, 20 and 45 min.

### 2.4. Mucin Penetration Assay

This experiment aimed to determine if the 5% (*v*/*v*) concentration of HerbELICO^®^ essential oil mixture could penetrate the 2.5% (*v*/*v*) artificial mucin barrier to inactivate the *H. pylori* cells beneath. As shown in Figure 3, there was a significant 38-fold reduction in the number of bacteria at one-hour postincubation, confirming the ability of the essential oils in penetrating the mucin barrier, while almost no *H. pylori* strains survived.

The untreated group (control) had no exposure to the HerbELICO^®^ essential oil mixture. The treated group had exposure to the 5% (*v*/*v*) HerbELICO^®^ essential oil mixture for 30 min, 1 h and 2 h incubation periods.

### 2.5. Customer Case Study

The consumers were taking HerbELICO^®^  *liquid* or *solid* dietary supplements without additional therapy for 45 consecutive days (daily dosage of 260 mg of HerbELICO^®^ essential oil mixture for 15 consecutive days, followed by 130 mg of the HerbELICO^®^ essential oil mixture for another 30 days).

Eight weeks after the end of the usage of the HerbELICO^®^
*liquid* or *solid* dietary supplements, the consumers were subjected to *H. pylori* stool antigen testing.

Six of the seven consumers who were taking HerbELICO^®^  *liquid* were *H. pylori* negative (eradication rate 86%), while all consumers (eight) taking HerbELICO^®^  *solid* had a negative *H. pylori* stool antigen test eight weeks after the end of the treatment (eradication rate 100%). Thus, the average eradication rate in this group was 93%. Ten consumers reported a complete loss of the symptoms (e.g., stomach pain, increased level of gastric acid, bloating) after 10 days from the beginning of the usage of HerbELICO^®^
*liquid* or *solid*, while the other five consumers lost the symptoms until the end of the treatment.

## 3. Discussion

Carvacrol [17,22] and thymol [17,21] are the major components of oregano and thyme essential oils and previously have shown in vitro anti-*H. pylori* activity. Additionally, some findings indicate that the mixture of oregano and savory essential oils is more effective against *H. pylori* growth compared to each oil being used individually [22]. Therefore, it is possible that these oils may have a synergistic effect when being used together [22].

Essential oils do not normally induce adverse side effects and are therefore generally recognized as safe by the U.S. Food and Drug Administration [23,24]. In addition, unlike antibiotics, it is probably very difficult for any micro-organism to develop resistance to essential oils [13,27].

Based on all the mentioned facts, we hypothesized that the mixture of oregano, savory and thyme essential oils, which was named the HerbELICO^®^ essential oil mixture in this manuscript, may be used as an alternative natural agent against *H. pylori* infection.

This study confirmed that the HerbELICO^®^ essential oil mixture is effective, even in 4-5% (*v*/*v*) concentrations (36.6–45.7 g/L), against 20 different *H. pylori* strains from different host origin and different resistance levels to antimicrobial medicinal products. This suggests that the HerbELICO^®^ essential oil mixture is effective against all *H. pylori* types regardless of their resistance or host origin. Since essential oils are complex mixtures of lipophilic compounds, they probably can penetrate through the bacterial cell wall, disrupt the bacterial membrane and interfere with integral membrane proteins, leading to cascade events of bacterial cell rupture [28]. In addition, many bacterial cellular functions can be affected: ion balance [14], energy conversion processes and depletion of the ATP [29], the synthesis of structural macromolecules [30], etc. Additionally, the chemical complexity of essential oils completely disables the microorganism from developing resistance. This implies that it is probably impossible for *H. pylori* to become resistant to the HerbELICO^®^ essential oil mixture.

Furthermore, the efficacy of the HerbELICO^®^ essential oil mixture was confirmed in two other in vitro tests: time-kill and mucin penetration assays. To the best of our knowledge, although a time-kill assay was applied to investigate the effectiveness of some antibiotics [31], it has never been used before to assess the efficacy of essential oils against *H. pylori.* Nevertheless, after 10 min of exposure to the HerbELICO^®^ mixture, all the tested *H. pylori* strains were killed. Then again, many natural products such as garlic [32] and manuka honey [33] have been shown in vitro to have a similar effect. Therefore, the next most important part of this study was to demonstrate the capability of the essential oils to penetrate through the mucus layer and reach the *H. pylori* colonizing underneath [34]. It was supposed that elevated pH, which is a consequence of the activity of *H. pylori* urease (urease transforms urea to ammonia and thus enables the survival of *H. pylori* in acidic conditions in the stomach), leads to changes in the rheology of gastric glycoprotein mucin (the main constituent of a mucus) from a gel to a viscous solution, enabling the mobility of *H. pylori* [35]. The mucus acts as a protective blanket for the stomach wall and also for *H. pylori.* The mucus prevents most of the exogenous substances from coming into direct contact with *H. pylori,* which explains why most of the natural products fail to eradicate *H. pylori* in vivo. This study demonstrated that the exposure of the HerbELICO^®^ essential oil mixture on top of the mucus for an hour is sufficient to kill off almost all *H. pylori* colonizing underneath. This can be rational evidence that it could reach not only the *H. pylori* found on the surface, but also the bacteria which are distributed in the weakened mucus gel. Additionally, the obtained results suggested that the developed pharmaceutical form should be taken orally on an empty stomach at least 30 min before a meal. This dosage regimen should be adequate to provide the possibility for essential oils to penetrate mucus and come in close contact with *H. pylori* beneath this layer.

Anyway, some further research on anti-*H. pylori activity* could include the major individual compounds found in the HerbELICO^®^ mixture (Table 1), where their dose-dependent and synergistic activities could be assessed.

In the customer case study, the consumers were taking two capsules of HerbELICO^®^
*liquid* and HerbELICO^®^  *solid* twice a day for 15 days and switched to one capsule twice a day for another 30 days. The difference was noticed between the cure rate in the subgroups taking HerbELICO^®^
*liquid* and HerbELICO^®^  *solid*: 86% and 100%, respectively, indicating that *solid* was more effective than *liquid*. However, this difference was based on only one consumer for which the treatment with HerbELICO^®^
*liquid* was not successful. Thus, the average cure rate for the *solid* and *liquid* groups together (93%) was considered.

The results of the consumer case study suggest that the applied dosage regiment (15 days two capsules twice a day + 30 days one capsule twice a day) was very successful in the eradication of *H. pylori* and was well tolerated with no reported adverse side effects. The loss of symptoms, no unwanted side effects and negative *H. pylori* test eight weeks after the end of treatment in 14 of the 15 consumers (eradication rate of 93%) confirmed that the HerbELICO^®^ dietary supplement in both *solid* and *liquid* forms has high potential in the eradication of *H. pylori* infection from gastric mucosa. Nevertheless, this customer case study, which aimed to collect initial information on the efficacy of the HerbELICO^®^ essential oil mixture, had certain limitations: the small number of participants, the nonrandom assignment and the absence of a placebo group. Thus, the conducted customer case study could not be used directly to make conclusions and advise a way of application of the HerbELICO^®^ dietary supplement in the treatment of *H. pylori* infection, but it could be used as a good starting point for further clinical trials.

## 4. Materials and Methods

### 4.1. HerbELICO^®^ Essential Oil Mixture and Pharmaceutical Dosage Forms

The HerbELICO^®^ essential oil mixture containing three essential oils isolated from *Satureja hortensis* L., *Origanum vulgare* L. and *Thymus vulgare* L. was supplied by HerbElixa Ltd. (Sremska Kamenica, Serbia; registration number: 21435252). The HerbELICO^®^ essential oil mixture is a homogeneous blend of savory, thyme and oregano oils that was created by adding various amounts of the individual oils to achieve the necessary concentrations of the main constituents (Table 1). The mixture was analyzed with GC-MS prior to further testing.

For the consumer case study, two pharmaceutical dosage forms of the HerbELICO^®^ essential oil mixture—HerbELICO^®^
*liquid* (registration number: 15173/2019) and HerbELICO^®^  *solid* dietary supplement (registration number: 16828/2020), which were available on the market—were used in this study. Both products contained 60 hard gelatine capsules. For the HerbELICO^®^
*liquid* product, each capsule contained 65 mg of the essential oil mixture and 435 mg of sunflower carrier oil. The HerbELICO^®^
*solid* capsule, on the other hand, contained 65 mg of the essential oil mixture adsorbed on solid carrier Florite^®^ (28 mg) and mixed with microcrystalline cellulose (214 mg) and magnesium stearate (3 mg).

### 4.2. GC-MS Analyses

A qualitative and semiquantitative chemical characterization of the HerbElLICO^®^ essential oil mixture was performed with GC-MS at the Department of Chemistry, Biochemistry and Environmental Protection, Faculty of Sciences, University of Novi Sad, Serbia, using Agilent Technologies 6890N gas chromatograph coupled with Agilent Technologies 5975B electron ionization mass-selective detector (Agilent Technologies, Santa Clara, CA, USA). An aliquot of 1 μL of essential oil dissolved in hexane (10 μL/mL) was injected into a split/splitless inlet at 250 °C with a split ratio of 1:10. Helium (purity 5.0) was used as a carrier, with a constant flow of 1 mL/min. The components were separated on a nonpolar Agilent Technologies HP-5 ms column (30 m × 0.25 mm, 0.25 μm) using the temperature program starting at 50 °C, increasing 8 °C/min to 120 °C, then 15 °C/min to 230 °C, and finally 20 °C/min to 270 °C, and holding at 270 °C for 16.9 min (total run time 35 min). The effluent was delivered to the mass spectrometer via a transfer line held at 280 °C. The ion source temperature was 230 °C, electron energy 70 eV and quadrupole temperature 150 °C.

To achieve a better correlation between the experimental and library spectra, a standard spectra tune was used. Data were acquired in scan mode (m/z range 35–400) with a solvent delay of 2.30 min. The data were processed using Agilent Technologies MSD ChemStation software (revision E01.01.335) combined with AMDIS (ver. 2.64) and NIST MS Search (ver. 2.0d). AMDIS was used for deconvolution, i.e., coeluting the compounds’ peak area determination and pure spectra extraction, and a NIST MS Search provided a search algorithm complementary to the PBM algorithm of ChemStation. The compounds were identified by a comparison of the mass spectra with data libraries (Wiley Registry of Mass Spectral Data, 7th ed. and NIST/EPA/NIH Mass SpectralLibrary 05) and confirmed by a comparison of the linear retention indices with the literature data [26]. Diesel oil, containing C8–C28 n-alkanes, was used as a standard for the determination of retention indices. Relative amounts of components, expressed in percentages, were calculated by a normalization procedure according to the peak area in the total ion chromatogram.

### 4.3. Bacterial Strains and Culturing

Twenty *H. pylori* clinical strains, as shown in Table 2 and Table 3, were used in this study. All cultures were routinely maintained on Columbia blood agar plates containing 5% horse blood for three to four days at 37 °C in a 10% CO_2_ environment.

### 4.4. Antimicrobial Medicinal Products Susceptibility Testing

Each *H. pylori* strain was tested for susceptibility against clarithromycin, ciprofloxacin, rifampicin, metronidazole, tetracycline and amoxicillin using the Etest^®^ strips (BioMérieux, Craponne, France). According to the MIC breakpoints provided in The European Committee on Antimicrobial Susceptibility Testing (EUCAST) guidelines, *H. pylori* is considered resistant if: >0.5 mg/L for clarithromycin, >1 mg/L for ciprofloxacin, tetracycline and rifampicin, >8 mg/L for metronidazole and >0.125 mg/L for amoxicillin.

### 4.5. Determination of the MIC of HerbELICO^®^ Essential Oil Mixture

The MIC of the HerbELICO^®^ essential oil mixture was determined using the agar dilution method on 24-well plates. In brief, the HerbELICO^®^ mixture dilutions were prepared in a brain heart infusion (BHI) agar supplemented with 10% (*v*/*v*, volume of reagents/total volume of reaction probe) heat-inactivated fetal calf serum (FCS) and 10% (*v*/*v*) dimethyl sulfoxide (DMSO). The final concentration of the HerbELICO^®^ mixture per tested well was in the range of 1–6% (*v*/*v*, volume of oil mixture/total volume of reaction probe), i.e., 9.15–54.9 g/L. For each *H. pylori* strain, grown for 48 h on blood agar plates, a bacterial suspension with an OD_600_ of 0.1 was prepared in phosphate buffered saline (PBS). Next, for the examination of bacterial growth, for each tested HerbELICO^®^ mixture concentration, 50 µL of bacterial suspension was added per well in triplicate, followed by incubating the plates at 37 °C in a 10% CO_2_ environment for 3 days. For each tested concentration, a positive control for bacterial growth (containing inoculum without HerbELICO^®^ mixture) and a negative control (containing HerbELICO^®^ mixture without inoculum) were included.

### 4.6. Time-Kill Assay

The killing activity of the 5% (*v*/*v*) HerbELICO^®^ essential oil mixture was evaluated against four *H. pylori* strains (HP14012, HP11055, HP11043 and HP15005) using 24-well plates. In each well, 2 mL of BHI broth containing 10% (*v*/*v*) heat-inactivated FCS, 0.5% (*v*/*v*) DMSO, 5% (*v*/*v*) HerbELICO^®^ mixture and bacterial cells standardized to a final OD_600_ of 0.1 were added. Bacterial inoculum-negative and HerbELICO^®^ mixture-negative wells served as the negative and positive controls, respectively. The plates were incubated at 37 °C in a 10% CO_2_ environment. At 10, 20 and 45 min postincubation, 0.5 mL of bacterial suspension was collected and centrifuged at 11,000× *g* for 1 min at room temperature. The supernatant was discarded prior to resuspending the bacterial pellet in 0.5 mL of saline. Dilutions of bacterial suspensions were plated on blood agar plates and incubated at 37 °C in a 10% CO_2_ environment for 3–4 days. Visible bacterial colonies were counted. The experiment was performed in triplicate and repeated on two separate occasions.

### 4.7. Mucin Penetration Assay

Mucin is the main constituent of gastric mucus, and these experimental conditions imitated a real situation in a human stomach infected with *H. pylori*. *H. pylori* colonize gastric epithelium under the mucus layers, which prevents exogenous substances from coming into contact with *H. pylori*.

The mucin chambers were prepared by transferring 200 μL of PBS containing 2.5% (*w*/*v*) porcine mucin (Sigma-Aldrich, St. Luis, MO, USA) and 0.3% (*w*/*v*) Bacto™ agar (Becton, Dickinson and Company, Franklin Lakes, NJ, USA) into individual 12 mm Transwell^®^ inserts with a 0.4 μm membrane pore size (Corning, Corning, NY, USA) in a 12-well plate. An hour later, 1 mL of the *H. pylori* strain HP15005 bacterial suspension, belonging to the clinical *H. pylori* strain collection of the Helicobacter Research Laboratory at The University of Western Australia, prepared by harvesting and resuspending a 48 h old plate culture in BHI broth containing 10% (*v*/*v*) heat-inactivated FCS to an OD_600_ of 0.1, was added to the bottom chamber. This was then followed by gently adding 500 μL of BHI solution containing the 5% (*v*/*v*) HerbELICO^®^ essential oil mixture, 0.5% (*v*/*v*) DMSO and 10% (*v*/*v*) heat-inactivated FCS to the top of the artificial mucin layers. The plate was incubated at 37 °C in a 10% CO_2_ environment. At 30 min, 1 h and 2 h postincubation, 100 μL of the bacterial suspension from the bottom chamber was collected. Following centrifugation at 11,000× *g* for 1 min at room temperature, the supernatant was discarded and the bacterial cells were resuspended in 100 μL of PBS. Dilutions of the bacterial suspension were plated on blood agar plates and incubated at 37 °C in a 10% CO_2_ environment for 3–4 days. Visible bacterial colonies were counted. The experiment was performed in triplicate and was repeated on two separate occasions.

### 4.8. Consumer Case Study

#### 4.8.1. Participants

A total of 15 consumers, aged between 18 to 75 years old, with *H. pylori* infection confirmed by stool antigen testing were recruited. The following exclusion criteria were applied: cardiovascular or renal disease, liver disease, cancer, pancreatitis, pregnancy/lactation, excessive alcohol intake or illegal drug use and intolerance to any ingredient present in HerbELICO^®^  *liquid*/HerbELICO^®^  *solid* dietary supplements. All participants signed an informed consent form. This study was approved by the Institutional Review Board at Poliklinika Consilium, Novi Sad, Serbia (Study approval 2020-1), where all consumers were interviewed by the medical doctor—a gastroenterologist.

#### 4.8.2. Procedure

During the study, the consumers were taking only the HerbELICO^®^  *liquid* or HerbELICO^®^  *solid* dietary supplement, but not any other therapy for *H. pylori* infection. The consumers were asked to have their normal dietary regime without any alcohol intake and to perform *H. pylori* stool antigen testing before and eight weeks after the treatment with HerbELICO^®^
*liquid* or HerbELICO^®^  *solid*. The testing was conducted using the *H. pylori* Antigen Rapid Test Cassette (Acro Biotech Inc, Rancho Cucamonga, CA, USA) by Polaris Laboratory in Novi Sad, Serbia.

Fifteen consumers with confirmed *H. pylori* infection and with mild symptoms of infection (e.g., bloating, stomach ache, reflux, coughing, loss of appetite) were included in this study. Since previous studies concluded that the spontaneous clearance rate is usually very low—2.9% per year [36]—the placebo group was excluded from this study, which was considered to be sufficient. Seven consumers were taking the HerbELICO^®^  *liquid* product and the remaining eight were taking the HerbELICO^®^  *solid* product. Each consumer began the treatment with four capsules per day, two in the morning and two in the evening on an empty stomach and 30 min before a meal, for 15 consecutive days, and then they switched to two capsules per day, one in the morning and one in the evening on an empty stomach and 30 min before a meal, for another 30 days. This corresponds to a daily intake of 260 mg of the HerbELICO^®^ essential oil mixture for 15 consecutive days, followed by 130 mg of the HerbELICO^®^ essential oil mixture for another 30 days.

The consumers were asked to answer the questions from two questionnaires during the consumer case study. The first questionnaire was designed to record the symptoms of *H. pylori* infection of each customer before treatment. The second questionnaire was designed to record the changes in the symptoms and possible side effects experienced by the consumer during treatment. Each consumer was interviewed by the medical doctor—a gastroenterologist—based on which questionnaires were filled in. Th eradication rate in each group was calculated as the number of consumers with a confirmed negative *H. pylori* stool antigen test eight weeks after the end of the treatment divided by the total number of consumers in the group.

### 4.9. Statistical Analysis

For statistical comparison, the paired Student *t-*test was employed. A *p-*value less than 0.05 was considered statistically significant.

## 5. Conclusions

The HerbELICO^®^ essential oil mixture is a homogenous blend of savory, oregano and thyme essential oils, characterized by the presence of carvacrol (47.44%), thymol (11.62%), *p*-cymene (13.35%) and *γ*-terpinene (18.20%) as the most dominant compounds. The efficacy of the HerbELICO^®^ essential oil mixture against *Helicobacter pylori* was confirmed in several assays, which directly supports its prospective in vivo activity. The HerbELICO^®^ essential oil mixture inhibited the growth of 20 different *H. pylori* types regardless of their resistance or host origin (MIC = 36.6–45.7 g/L; 4–5% *v*/*v*), thus showing nonselective inhibitory activity. It also expressed immediate bactericidal activity (10 min exposure to HerbELICO^®^ was enough to kill off the examined *H. pylori* strains) and the ability to penetrate through mucin. In the consumer case study, both pharmaceutical dosage forms—HerbELICO^®^  *liquid* and *solid* dietary supplements—had high acceptance by consumers and high eradication rates. Bearing in mind the limitations of the conducted customer case study, such as a small number of participants, a bigger cohort is needed to further evaluate the efficacy of such treatment protocol.

## 6. Patents

The details of the chemical composition of the HerbELICO^®^ essential oil mixture are a subject of the patent application NO. PCT/RS2020/000016.

## Figures and Tables

**Figure 1 molecules-28-02138-f001:**
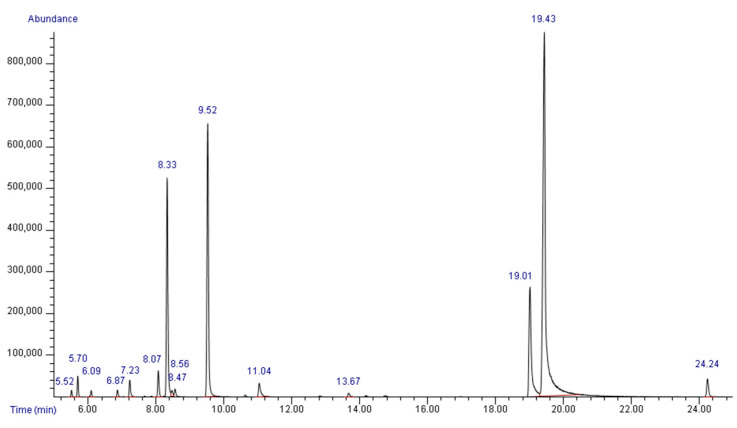
GC-MS chromatogram of HerbELICO^®^ essential oil mixture.

**Figure 2 molecules-28-02138-f002:**
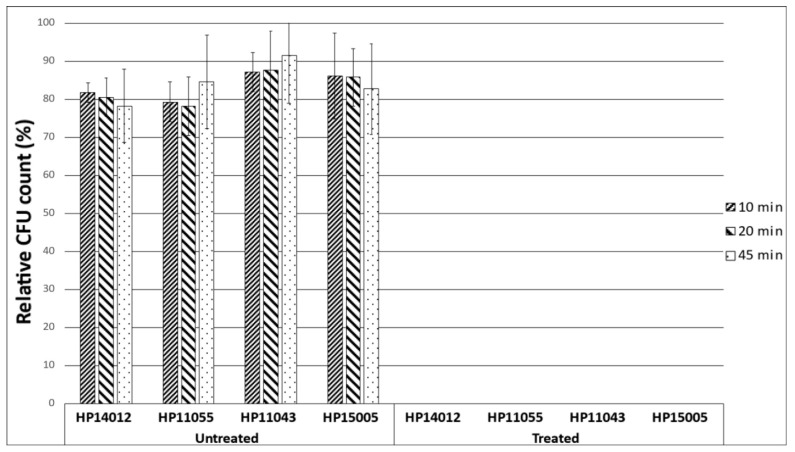
Time required for HerbELICO^®^ essential oils mixture to kill off *H. pylori* cells.

**Figure 3 molecules-28-02138-f003:**
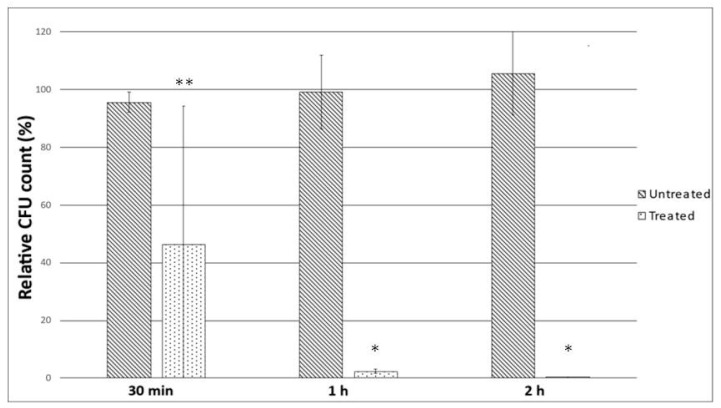
Results of mucin penetration assay showing time required for HerbELICO^®^ essential oils mixture to penetrate mucin layer and kill off *H. pylori* cells beneath. * statistically different from untreated group (control) (*p* < 0.01); ** statistically not different from untreated group (control) (*p* > 0.01).

**Table 1 molecules-28-02138-t001:** The main constituents (% of total peak area) of HerbELICO essential oil mixture determined by GC-MS.

AI_exp_	AI_lit_	Compound	% of Total Peak Area
925	924	*α*-Thujene	0.32
932	932	*α*-Pinene	0.99
947	946	Camphene	0.37
976	974	*β*-Pinene	0.36
990	988	*β*-Myrcene	0.96
1005	1002	*α*-Phellandrene	0.05
1016	1014	*α*-Terpinen	1.51
1023	1020	p-Cymene	13.35
1027	1024	Limonene	0.42
1030	1026	1,8-Cineole	0.52
1057	1054	*γ*-terpinene	18.20
1088	1086	*α*-Terpinolene	0.13
1100	1095	Linalool	1.45
1141	1141	Camphor	0.10
1164	1165	Borneol	0.35
1174	1174	Terpinen-4-ol	0.12
1186	1186	*α*-Terpineol	0.19
1292	1289	Thymol	11.62
1302	1298	Carvacrol	47.44
1418	1417	*β*-Caryophyllene	1.56

AI_exp_—arithmetic retention index obtained in the experiment; AI_lit_—arithmetic retention index from the literature [26].

**Table 2 molecules-28-02138-t002:** The MIC values of antimicrobial medicinal products for *H. pylori* strains, obtained by E-test.

Strain	Patient Origin	MIC (mg/L)					
		MTZ	AMX	CIP	TET	CLA	RIF
HP08058	Sudan	256 ^†^	0.016	32 ^†^	0.023	8 ^†^	1.5 ^†^
HP11043	Afghanistan	256 ^†^	0.016	0.064	0.125	8 ^†^	32 ^†^
HP11049	Sudan	32 ^†^	0.016	0.047	0.016	12 ^†^	0.38
HP11055	Vietnam	256 ^†^	0.016	0.125	0.023	256 ^†^	0.25
HP13050	Sudan	0.25	0.016	0.094	0.016	0.016	0.38
HP13064	Iran	256 ^†^	0.016	0.5	0.016	256 ^†^	2 ^†^
HP14012	Vietnam	0.064	0.016	32 ^†^	0.125	256 ^†^	2 ^†^
HP15005	Eritrea	256 ^†^	0.016	0.047	0.016	0.016	0.38
HP15026	Afghanistan	256 ^†^	0.016	32 ^†^	0.016	8 ^†^	0.38
HP15035	Iran	256 ^†^	0.19 ^†^	0.032	0.5	1 ^†^	0.125
HP15065	India	256 ^†^	0.016	32 ^†^	0.016	256 ^†^	0.25
HP16024	Vietnam	0.75	0.125	32 ^†^	0.047	256 ^†^	1.5 ^†^
HP16033	Bangladesh	256 ^†^	0.125	0.016	0.016	32 ^†^	0.094
HP16035	Vietnam	256 ^†^	0.016	0.008	0.032	2 ^†^	0.38
HP16037	India	0.38	0.016	3 ^†^	0.064	0.032	1
HP16046	Vietnam	0.25	0.125	0.032	0.38	256 ^†^	0.25
HP16051	Vietnam	256 ^†^	0.016	0.032	0.032	32 ^†^	3 ^†^
HP17023	Lebanon	0.25	0.047	0.047	0.047	0.094	2
HP17026	Eritrea	256 ^†^	0.016	0.032	0.016	256 ^†^	0.25
HP17029	Mauritius	0.5	0.016	0.032	0.016	0.016	0.5

^†^—Antimicrobial medicinal products resistance. Abbreviations: MIC: minimum inhibitory concentration; MTZ: metronidazole; AMX: amoxicillin; CIP: ciprofloxacin; TET: tetracycline; CLA: clarithromycin; RIF: rifampicin.

**Table 3 molecules-28-02138-t003:** The MIC values of HerbELICO^®^ essential oil mixture for *H. pylori* strains obtained by agar dilution method.

Strain	HerbELICO^®^ Essential Oil Mixture	
	MIC (*v*/*v* %)	MIC (g/L)
HP08058	4	36.6
HP11043	5	45.7
HP11049	4	36.6
HP11055	5	45.7
HP13050	4	36.6
HP13064	5	45.7
HP14012	5	45.7
HP15005	4	36.6
HP15026	4	36.6
HP15035	5	45.7
HP15065	5	45.7
HP16024	5	45.7
HP16033	5	45.7
HP16035	5	45.7
HP16037	5	45.7
HP16046	5	45.7
HP16051	4	36.6
HP17023	4	36.6
HP17026	5	45.7
HP17029	5	45.7

## Data Availability

All data supporting the reported results are shown in the manuscript.

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
