# Peer review of "Savory, Oregano and Thyme Essential Oil Mixture (HerbELICO^®^) Counteracts *Helicobacter pylori"

_molecules, 2023, doi:10.3390/molecules28052138_

Round 1

Reviewer 1 Report

As recorded on the attached file.

Author Response

Thank you for your suggestions, all of them are accepted and all changes were made.

Reviewer 2 Report

I herebin accept the article and suggest a complementry work to be done using the individual major components as the authors plane to do in future 

Author Response

I herebin accept the article and suggest a complementry work to be done using the individual major components as the authors plane to do in future.

Thank you for your suggestion, some further experiments are ongoing, and we will certainly include individual major components and their combinations, in order to assess their dose-dependent and synergistic activities.

Reviewer 3 Report

In view of the current chemical drugs in the treatment of Helicobacter pylori at the same time produced adverse side effects. The authors used a variety of essential oil mixtures (named HerbELICOÂ ® essential oil mixtures) for the treatment of Helicobacter pylori infection. The research has some value and will interest readers of the journal.

1.     Line 62-74, the specific activity of natural product essential oil is not clear enough. What is the specific MIC value? In addition, 3-5 literature reports should also be given on the related research of compound essential oil.

2.     “HerbELICO® essential oil mixture” You may consider providing a physical map, or a sample map of your product.

3.     Line 247-249you use the data libraries, perhaps you should put the matching results as a table. In addition, by comparison of linear retention indices with literature data, you determine the structure of the compound, and should refer to the corresponding literature annotation in table 1, so that the reader can understand the source of the literature reference. In general, mass spectrometry data of compounds should be listed in table, compared with the Mass spectrometry data in the database. The Greek letters in Table1 need to use italics.

4.     Line 80-86, the author analyzed the chemical structure. Has the author considered a question ' the interaction between polyphenols and essential oils ', essential oils can be used as a solvent for polyphenols, and how do they work anti-Helicobacter pylori?

5.     Line 97-98, In general, the MIC value of antibacterial is used in the mass volume unit of g / L. Does the author have relevant data ? Why use v/v ?

6.     In Table2, the positive drug MIC value is mg / L, while HerbELICO uses the unit of v / v. How to compare ?

7.     In Figure 3, what is the error of 30 min treated bar (Mean±SD)? If the P value is less than 0.01, it should be expressed as three stars. Please verify. It is suggested that figure 2 and 3 be redrawn.

Author Response

In view of the current chemical drugs in the treatment of Helicobacter pylori at the same time produced adverse side effects. The authors used a variety of essential oil mixtures (named HerbELICO® essential oil mixtures) for the treatment of Helicobacter pylori infection. The research has some value and will interest readers of the journal.

Thank you for all your suggestions, changes are accepted, and explanations are given below.

  1. Line 62-74, the specific activity of natural product essential oil is not clear enough. What is the specific MIC value? In addition, 3-5 literature reports should also be given on the related research of compound essential oil.

Specific MIC or MBC values for some relevant examples are added. But, to the best of our knowledge, only in Lesjak et al. (2015) research of compound essential oils (mixture of several essential oils) and their influence on H. pylori was undertaken.

  1. “HerbELICO® essential oil mixture” You may consider providing a physical map, or a sample map of your product.

More detailed description is given in Materials and Methods (4.1. HerbELICO® essential oil mixture and pharmaceutical dosage forms).

  1. Line 247-249,you use the data libraries, perhaps you should put the matching results as a table. In addition, by comparison of linear retention indices with literature data, you determine the structure of the compound, and should refer to the corresponding literature annotation in table 1, so that the reader can understand the source of the literature reference.

Arithmetic retention indices from literature are added in Table 1, as well as reference [26] (Adams, 2012).

In general, mass spectrometry data of compounds should be listed in table, compared with the Mass spectrometry data in the database.

In general yes, but in the field of GC/MS analysis of essential oils the book [26] Adams, 2012:”Identification of Essential Oil Components by Gas Chromatography/Mass Spectrometry” is a standard library and there is no need for listing MS spectral data in a table.

The Greek letters in Table1 need to use italics.

Changed.

  1. Line 80-86, the author analyzed the chemical structure. Has the author considered a question ' the interaction between polyphenols and essential oils ', essential oils can be used as a solvent for polyphenols, and how do they work anti-Helicobacter pylori?

No, we did not considered 'the interaction between polyphenols and essential oils', since we were focused only on activity of essential oils, which constituents are volatile terpenoids, while polyphenols usually are not volatile and are  not extracted during the isolation of essential oils (hydrodistillation of plant material). We found this suggestion more than interesting, and we will include assessment of possible interaction between plant polyphenols and terpenoids in our further experiments.

  1. Line 97-98, In general, the MIC value of antibacterial is used in the mass volume unit of g / L. Does the author have relevant data ? Why use v/v ?

Since essential oils are liquids it is common to express their concentration as v/v. The density of mixture was 0.915 g/mL so v/v can be converted to g/L.

  1. In Table2, the positive drug MIC value is mg / L, while HerbELICO uses the unit of v / v. How to compare ?

MIC for antibiotics is determined by E test, and for essential oil by agar dilution method. Thus, the results are not comparable. The test for antibiotics is only used to confirm that most of the strains are antibiotic resistant.

In order to avoid confusion, the results for antibiotics and essential oils are separated into tables - Table 2 and Table 3, and in Table 3 column with MICs  expressed in g/L is added.

  1. In Figure 3, what is the error of 30 min treated bar (Mean±SD)? If the P value is less than 0.01, it should be expressed as three stars. Please verify. It is suggested that figure 2 and 3 be redrawn.

In Figure 3, there is no statistical difference in 30 min treatment. The stars are added.

Round 2

Reviewer 3 Report

The authors have corrected these errors. So, I surpport this paper's publication in MOLECULES.

Author Response

Thank you!